# Developing an Acceptable Nixtamalised Maize Product for South African Consumers: Sensory, Survey and Nutrient Analysis

**DOI:** 10.3390/foods13182896

**Published:** 2024-09-12

**Authors:** Taylon Colbert, Carina Bothma, Wilben Pretorius, Alba du Toit

**Affiliations:** Department of Sustainable Food Systems and Development, Faculty of Natural and Agricultural Sciences, University of the Free State, P.O. Box 339, Bloemfontein 9300, South Africa; colbertta@ufs.ac.za (T.C.); bothmac@ufs.ac.za (C.B.); pretoriusw1@ufs.ac.za (W.P.)

**Keywords:** consumer perception, Just-About-Right (JAR), consumer’s overall liking, vegetarian nuggets and burgers, nixtamalised maize chip, home-based production

## Abstract

South Africa produces high-quality maize, yet food insecurity and malnutrition are prevalent. Maize is a staple for most South Africans and is often eaten as pap, gruel cooked from maize meal (corn flour) and water without diet diversification. Considering the reliance on maize in low-income communities, could nixtamalised maize products be developed that are nutritious, homemade and consumer-acceptable? Nixtamalisation could offer a solution. However, its acceptability and nutritional benefits remain in question. We aimed to develop a product using consumer-led methods. Consumer panels evaluated and selected products using overall acceptability (9-point hedonic scale), Just-About-Right (JAR) and penalty analysis. Consumer-acceptable nixtamalised chutney-flavoured maize chips were moderately liked (7.35) and reached acceptable JAR scores (74.2%). The nixtamalised products were liked and liked very much (56%), 61% of panel members agreed and strongly agreed to purchase and prepare, and 50% to consume nixtamalised products. Nutrient analysis of the chutney chips showed high energy (2302 kJ/100 g) and total fats (23.72), of which saturated fats were 11.47%. Total fibre (17.19 g/100 g), protein (6.64 g/100 g), calcium (163.3) and magnesium (53.67 g/100 g) were promising, while high phosphorous (566.00 mg/100 g) may indicate anti-nutrients present. Nixtamalisation can alleviate food insecurity and malnutrition in countries such as South Africa.

## 1. Introduction

Globally, maize (*Zea mays*, L.), also known as corn, contributes significantly to human and animal diets and thus plays a pivotal role in food security [1]. Food security is defined as a situation that exists when all people, at all times, have physical, social, and economic access to sufficient, safe, and nutritious food that meets their dietary needs and food preferences for an active and healthy life [2]. Therefore, household food security exists when food is available, accessible, and affordable to households [3]. However, household food insecurity is the leading cause of malnutrition and is directly or indirectly caused by inadequate food consumption and poor diet quality [4].

Consuming a consistently diversified diet poses a challenge, as most African populations consume monotonous starchy staples coupled with a limited intake of animal products, fruit, and vegetables [5]. A diversified diet consists of protein sources, such as beans, peas, fish, meat, and milk, as well as fruits and vegetables, which are abundant in vitamins and minerals, along with carbohydrates from cereals, which can effectively mitigate the challenges associated with malnutrition [6]. Bouis (2003) emphasised that addressing malnutrition involves consuming non-staple foods, particularly animal products, which are abundant in bioavailable micronutrients [7]. However, due to financial constraints, many households in developing countries need help to consume a nutritionally balanced diet [7]. Lower-income groups are especially susceptible to malnutrition-related diseases and deaths due to the inaccessibility of affordable, nutrient-dense foods [8]. Drammeh et al. (2019) further indicate that inadequate nutrition can impair physical and mental development and reduce productivity among children under the age of five and throughout their lives [4].

South Africa is generally considered food secure because when the agricultural sector cannot produce sufficient staple crops, they can be imported [9]. However, researchers believe that this is not true at the household level [10,11]. Inequality faced by South African citizens due to Apartheid has resulted in high levels of unemployment and poverty among non-white South Africans. Adverse conditions existed throughout the regime, from its inception (1948) to its eventual dismantling in 1994 [12]. Inequality persists in South Africa, and according to StatsSA (2021), approximately 20% of South African households had inadequate access to food in 2017, and more black and coloured households face food insecurity when compared to white households [13].

And yet, South Africa is well known for its reputable maize production. Approximately eight million tonnes of maize grain are produced annually in South Africa, with the Free State, North West, Mpumalanga, and KwaZulu-Natal being the major maize-producing provinces. Collectively, these provinces account for about 83% of the total national production of maize [13]. It serves as a staple for most of the population and is a significant component of animal feed [13]. Maize and maize-based products are consumed by approximately 67% to 83% of the population. The daily consumption averages between 476 g and 690 g per person [14].

Thus, the significance of maize in the food industry cannot be understated, as maize meal products are consumed almost exclusively as pap (gruel cooked from maize meal and water) [15]. Govender (2014) identified the consumption of pap as a significant problem, as it does not provide enough nutrients to nourish the body [16]. Many people highly prefer to consume pap, particularly in lower-income and rural communities [17]. This predominant reliance on maize and maize-based dishes (pap) without diversification in the diet exaggerates the issues of food and nutrition insecurity [6]. Diets often need to be more balanced in the supply of essential nutrients [6]. Maize is very high in carbohydrates but lacks several essential nutrients, such as the amino acids tryptophan and lysine [18].

It is also partly due to the presence of resistant starch phytate in maize, which acts as an anti-nutritional factor by binding to essential nutrients present in maize, consequently hindering the availability for digestion and absorption in the body [19]. Furthermore, conventional processing techniques involve the removal of the pericarp and germ, leading to the loss of critical nutrients, such as fibre, lipids, minerals, vitamins, and essential amino acids [20].

Given the difficulties in achieving a diversified diet among most populations in developing countries, countries such as South Africa, Zimbabwe, Nigeria, Malawi, Uganda, and Kenya have implemented mandatory food fortification. These countries fortify essential commodities, such as salt, bread, maize meal, wheat flour, and infant formulas with various vitamins and minerals [21]. However, food fortification presents its own set of limitations, such as mineral toxicity, due to inefficient monitoring systems by processors [21,22]. In addition, extra costs are incurred by the processor, which are included in the price of the final product [23]. Therefore, fortified foods are more costly than unfortified foods. Furthermore, relying on fortified foods to address nutrient deficiencies has limited coverage in rural areas, as most rural populations depend on home-based food products that are not fortified [22]. Despite government efforts, South Africa was listed as one of the 20 countries with severe child food poverty, meaning that children are fed only two or fewer food groups per day [24].

There is a need for cost-effective alternative processing techniques for maize that can improve the nutrient composition of maize-based products, particularly considering their reliance on this staple food. An old but practical approach to processing maize that has the potential to increase its nutritional value without specialised equipment is nixtamalisation, which is a traditional processing technique used in Mexico, but not widely recognised in Africa [25].

Nixtamalisation is not only a well-known industrial processing method but can be practised by any consumer who has access to whole-dried maize kernels and basic kitchen utensils. It is a process that could play an important role in decreasing food insecurity and malnutrition in poorer rural communities [26].

The nixtamalisation process involves cooking maize kernels in an alkaline-aqueous solution of calcium hydroxide (slaked lime), followed by soaking for between eight and 24 h. After that, the soaked kernels are thoroughly rinsed to remove pericarp and residual alkali, known as nejayote [27]. After rinsing, maize is called nixtamal, which can be ground to form a pliable dough, known as masa, or dried and ground into maize meal or flour [28]. The process has many nutritional benefits that the maize kernel lacks, such as increased niacin bioavailability, which decreases the risk of pellagra; higher calcium intake as a result of the steeping process with calcium hydroxide; increased dietary fibre by elevating the content of resistant starch; a significant reduction in mycotoxin levels; and diminished levels of phytic acid, which inhibits the bioavailability of calcium, iron, and zinc [29,30]. Additionally, nixtamalisation increases the essential amino acids lysine and tryptophane [31]. Therefore, combining this method with conventional processing techniques in Africa increases the nutrient content and reduces mycotoxin contamination, which is still a serious concern and poses a barrier to achieving food and nutrition security [25,32,33].

It is unclear why nixtamalisation as a food processing technique has not been introduced in South Africa when it holds so many advantages for the consumer. It is widely known that maize is often grown in backyards and community plots as part of or in addition to vegetable gardens. Using nixtamalisation, home-grown dried whole maize kernels could be converted into safe and delicious meals in homes using basic equipment, as it is widely and effectively practised in Mexico by rural women [34].

However, it needs to be determined if consumers would accept nixtamalised maize products that could be produced from home-grown whole-dried maize kernels and if the nutritional value would be comparable to commercial products. The aim of this study was to develop a consumer-acceptable nixtamalised maize product by employing consumer-led methods that could be produced in a household kitchen from whole-dried maize kernels.

To achieve this goal, nixtamalised maize was used to develop three well known and traditionally accepted products: vegetarian nixtamalised maize-based nuggets, burgers, and chips. To involve consumers in product development, consumer panels participated in sensory evaluations using the overall acceptability (hedonic scaling) and just-about-right (JAR) scales. Using a rigorous approach, the products were tested against similar products to provide reference products for comparison [35]. The best product was developed further and reformulated to achieve a commercial quality product that could be produced at home using dried maize kernels.

A survey was conducted to assess the acceptability of the four attributes (appearance, aroma, taste, and texture) as well as the willingness to repeat consumption, purchase, and recommend the products. The last section included questions to assess the conditions under which nixtamalised products would be consumed and purchased. Lastly, the nutrient content was determined and compared to the nutrient content of similar products.

## 2. Materials and Methods

### 2.1. Sample Preparation

The maize used in this study was received as dry white maize kernels from a farm near the small town of Hoopstad in the Lejweleputswa district of the Free State Province. The kernels were stored in airtight containers in a refrigerator at 4 °C in the Food Laboratory at the University of the Free State (UFS), Bloemfontein. White maize is the most commonly cultivated maize for human consumption in South Africa [36].

### 2.2. Description of Products

Nuggets are battered and breaded products that are popular in American fast-food restaurants. The most common nugget type is chicken nuggets, which are chicken breasts coated in a batter, rolled in breadcrumbs and fried in oil until cooked through [37]. As products are benchmarked against similar products in sensory testing, the maize nugget was tested in combination with two other vegetarian nuggets: chickpea [38] and potato [39].

Characteristically, burgers are meat products made from ground beef that may or may not contain additional ingredients [40]. The vegetarian maize burger was benchmarked using two vegetarian burger samples: a black bean [41] and a chickpea [42].

Chips (crisps) are thin slices of potatoes that are deep-fried until they are crunchy and flavoured with seasoning. These chips are commonly eaten as snacks [43]. Additionally, a wide variety of chips are available in the market. The benefits of chips produced from maize instead of potatoes are the vitamin and mineral contents. However, three maize chip samples were tested and formulated similarly using different cooking methods: one sample was deep-fried in oil, and two samples were air-fried at different temperatures and intervals. The chips were deep-fried because of the distinctive taste and texture qualities they impart to fried food [44]. However, deep-fried foods often contain high-fat content, comprising almost as much as 35–44% of the total product by weight [45]. Hence, the two samples were air-fried to provide a healthier option for health-conscious consumers. Air frying uses hot air to cook food, eliminating the need for excessive amounts of oil [46]. Consumers are becoming increasingly aware of the advantages of low-fat and fat-free products that do not require deep-frying [45]. This cooking method results in a reduced fat content of up to 80%, providing a healthier option for health-conscious consumers who still want to enjoy snack food [47]. It was further noted that lower oil content, such as crust formation, palatability, and appearance, may influence many sensory properties [45].

### 2.3. Nixtamalisation

The traditional nixtamalisation process was employed in this study since it is a simple and economical method practised in a home kitchen using basic kitchen equipment. Dry maize kernels were used as they are a very economical product that households can easily access.

Nixtamalisation was carried out in a 7-step process [33,48,49]. The ingredients used were six cups (1080 g) of whole dry maize kernels, 4.5 litres of water, and 4.5 teaspoons (10.4 g) of slaked lime. The ingredients were added to a stainless-steel pot and brought to a boil for 10 min until the pericarp loosened. After that, the maize kernels were left to soak overnight. The following day, the maize kernels were rubbed between gloved hands to remove the pericarp. The maize was transferred to a colander and rinsed under cold running water until the water was clear. The maize is now called nixtamal. The nixtamal was placed in a Kenwood Everyday Food Processor (FDP03.A0W) and finely ground to produce masa. The masa was spread evenly on baking sheets, lined with baking paper, and placed in an oven set at 100 °C, with the door slightly open, for 120 min. After that, the dried masa was then ground into a fine powder in a Nutribullet 600 W High-Speed Blender. The fine powder was sieved (0.8 mm) to remove unrefined particles. This process resulted in the production of masa flour. One cup (250 mL) of masa flour weighed an average of 165 g. This masa flour was used in the formulation of all the products.

To summarise, maize nuggets were evaluated with potato and chickpea nuggets, maize burgers with black bean and chickpea burgers, and maize chips were prepared in three ways (air-fried 1, air-fried two, and deep-fried). The formulations of the nixtamalised maize-based nuggets, burgers, original chips, and flavoured chips are regarded as the intellectual property of the UFS and are, therefore, not published.

### 2.4. Population Sampling and Panel Description

The target population was consumers; thus, naïve consumers who participated in the sensory evaluations were students and staff between 18 and 65 years of age from the University of the Free State on the Bloemfontein campus. The demographics of the four panels are shown in Table 1. It was affective testing that took place in the institution’s sensory laboratory, in a central location (CLT). Posters that advertised the tasting panel were designed to recruit consumers who regularly ate maize. The panels comprised volunteers who completed a consent form containing information on the study’s purpose and details concerning potential product allergens. Participation was anonymous, and no one with related food allergies was part of the study.

Four sensory panels were conducted [panel 1: nuggets, *n* = 79 (65 women and 14 men); panel 2: burgers, *n* = 83 (69 women and 14 men); panel 3: chips, *n* = 74 (61 women and 13 men); and panel 4: flavoured chips, *n* = 99 (83 women, 14 men, and two others). Before each panel, the panellists were trained to rank products using the overall liking and JAR method. The nuggets and burgers were prepared two hours before sensory tasting. Before transferring to the tasting trays, all samples were cut into 3 cm × 3 cm cubes and served warm in small tasting bowls covered with aluminium foil. For the chip panels, the chips were prepared a maximum of five days ahead, and three chips measuring 4 cm in diameter were used at each tasting. The panellists could not identify the samples as they were three-digit numerical codes. The samples were served on a white polystyrene tray under white light to each panel member seated in individual sensory booths, with no means to identify the samples. Room temperature water, ranging from 20 to 22 °C, was provided to cleanse the pallet between each sample [50] and a spitting cup for the less-than-satisfactory samples. The respondents received a small reward for participating in the study.

### 2.5. Sensory Analysis

#### 2.5.1. Consumer Liking Rating (Hedonic Evaluation)

Firstly, the products were evaluated using a consumer liking rating using the overall liking 9-point hedonic scale [51]. The most widely used scale for measuring food acceptability is the overall liking, a nine-point hedonic scale. It was administered at the same time as the JAR scaling task. The nine-point hedonic scale used in the consumer’s overall liking task consists of nine categories: 1 (dislike extremely), 2 (dislike very much), 3 (dislike moderately), 4 (dislike slightly), 5 (neither like nor dislike), 6 (like slightly), 7 (like moderately), 8 (like very much), and 9 (like extremely). The panellists were asked to taste the sample and rank it on a nine-point scale by marking one of the categories [52].

#### 2.5.2. Just-about-Right Evaluation

Secondly, the panellists were expected to complete the JAR scales, as shown in Table 2, for four attributes, namely aroma, appearance, taste, and mouthfeel [52]. The JAR scale is a bipolar assessment with two opposite reference points labelled “not enough” and “too much”, while the midpoint is labelled “just-about-right” (JAR), which the respondents used to rate the samples on the five-point scale [53]. The respondents were asked to tick a box on a five-point JAR scale on a ballot sheet (much too low, somewhat not enough, just-about-right, somewhat too much, and much too high). Each sample was evaluated for four attributes: appearance, aroma, mouthfeel, and taste. The analysis of the JAR scales treats all attributes as independent of one another. It is possible for the adaptation of one attribute to have a positive effect on other attributes, thereby eliminating the need to make changes to all attributes [54]. The JAR scale results in the study were obtained by counting the number of panel members who selected each category and calculating the corresponding percentage for each category. The scores for “much too low” and “somewhat too low” were combined and denoted “not enough”. Similarly, scores for “somewhat too high” and “much too high”) were combined and denoted “too much” [55]. The scores were combined to streamline the results; instead of displaying five results for the JAR scale, only three were presented [55]. The benchmark for good performance was set at 75% JAR for each attribute [35].

#### 2.5.3. Penalty Analysis

Data from JAR and consumers’ overall liking tasks were needed to determine the penalty analysis [56]. The penalty shows the link between the results of the consumers’ liking scores and the JAR scores. It indicates that respondents who were unsatisfied with the JAR test were also unsatisfied with the consumer’s overall liking task and vice versa [56]. It identifies an improvement opportunity that may bring the product closer to the ideal and shows the strength of prioritising improvement among attributes [35]. Penalty analysis is inconsequential (N/A) when the non-JAR score is less than 20.0, therefore indicating that less than 20% of respondents scored the attribute “not enough” or “too much” [35]. The penalty analysis has two results that give the product developer an indication of the attributes that have the greatest potential and could be improved. The mean drop is a measure of the impact of the “not enough” and “too much” scores (thus, each attribute has two mean drop penalties). The higher the mean drop, the greater the negative effect and 1.0 was considered the threshold. The penalty value also accounts for the number of respondents who gave the same score. It is used to determine whether changing a particular attribute through reformulation would impact the ranking of the attribute. An upper benchmark of 1.0 for the penalty was set as a threshold before an attribute should be improved or reformulation is necessary. The *p*-value for the penalty is calculated using ANOVA. It compares the JAR level’s mean with that of the other levels’ mean. This is equivalent to testing whether the penalty is significantly different from 0 or not.

### 2.6. Survey

After evaluating the products, the panel members were asked to complete a survey. It was requested that panel members complete the questionnaire only once. Thus, the questionnaire was never completed more than once by the same panel members, even though the same panel members might have taken part in more than one panel (*n* = 186) (145 women, 32 men, one other, and 8 chose not to disclose gender).

The questionnaire was designed to conduct exploratory quantitative research. The respondents were asked to respond in terms of their overall perception of nixtamalised maize products rather than for one specific product. Each respondent was allocated 15 min to taste the products before completing the questionnaire, which took approximately five minutes.

Section one of the questionnaire contained questions to determine the demographic profile of the respondents regarding age and gender. Section two included questions to assess the acceptability of the four attributes (appearance, aroma, taste, and texture) and questions to determine the willingness to repeat consumption, purchase, and recommendation of the products. The last section included questions to assess the conditions under which nixtamalised products would be consumed and purchased.

All internal consistency estimates, as determined by Cronbach’s alpha coefficient, exceeded 0.80, rendering the data reliable, and confirming that the individual items of a construct were consistently measured within the same construct or concept.

### 2.7. Determination of the Nutrient Content

As the chutney-flavoured chips were considered the most successful product, three batches were produced, analysed (in triplicate) for nutrient content, and compared to two commercial maize chips. The gross energy values of the chip were measured in megajoules (MJ) using a bomb calorimeter and converted to kJ/100 g [57].

#### 2.7.1. Determination of the Crude Protein Content

For crude protein determination, the Dumas method of combustion (ASM 056) [58] with a LECO FP 2000 machine (LECO Corporation, St. Joseph, MI, USA) was utilised [59]. The analysis was carried out by weighing 1 g of oven-dried sample and placing it directly into a reusable boat, which was then placed in the purge chamber of a horizontal furnace. The sample was combusted in the presence of oxygen at a temperature of 950 °C to determine its nitrogen content. The crude protein of the chips (g/100 g DM) was calculated by multiplying the nitrogen content (g/100 g DM) by a factor of 6.25 [60].

#### 2.7.2. Determination of the Fibre Content

The acid detergent fibre (ADF) method (ASM 059) was used to determine the fibre content present in the chips [58,61]. It involved dissolving 20 g N-cetyl-N, N, N-trimethyl ammonium bromide (C19H42BrN) in 1 litre (L) 1 N sulphuric acid (28 mL 98% sulphuric acid filled up to 1 L with distilled water) in glass pill vials. It was dried overnight in an oven at 100 °C for 24 h, placed in a desiccator for 30 min to reach room temperature, placed in a hot extraction unit, and cooled in a condenser. One hundred millilitres of acid detergent solution (ADS) was added to the samples and then heated on an element for 60 min. The samples were filtered, washed three times with hot distilled water, and rinsed twice with acetone. Afterwards, the samples were oven-dried overnight at 100 °C and cooled in a desiccator for 30 min. Afterwards, the samples were placed in a muffle furnace at 550 °C for four hours and then cooled for 30 min. The remaining ash was used to determine the fibre content [61].

The neural detergent fibre (NDF) was determined by the procedure outlined by Van Soest, Robertson and Lewis (1991) [62]. The sample preparation and analysis involved multiple steps, including oven drying at 55 °C, grinding, and 0.5 g of the sample placed in a 600 mL Berzelius beaker. Approximately 45 mL of NDF solution was added to a digestion burette and heated. The sample was mixed and then weighed into a plastic weigh pan. After gentle boiling, the sample was poured into a burette and refluxing was continued for 60 min. Afterwards, 2 mL of amylase solution was added, and the burette sides were rinsed. Glass crucibles were hotly weighed before filtration, and the samples were filtered using a vacuum. Acetone was added, and the samples were rinsed before being placed in a 105 °C oven overnight. The samples were weighed using crucibles the following day.

#### 2.7.3. Determination of the Fat Content

Total lipids from the maize chips were quantitatively extracted [63] using chloroform and methanol in a ratio of 2:1. An antioxidant, butylated hydroxytoluene, was added at a concentration of 0.001% to the chloroform−methanol mixture. A rotary evaporator dried the fat extracts under vacuum overnight at 50 °C using phosphorus pentoxide as a moisture adsorbent. Total extractable fat was determined gravimetrically from the extracted fat and expressed as percentage fat (*w*/*w*) per 100 g of the samples. The fat extracted from the maize chips was stored in a polytope (glass vial, with a push-in top) under a nitrogen blanket and frozen at −20 °C, pending the fatty acid analyses.

A lipid aliquot (25 mg) was transferred to a Teflon-lined screw-top test tube using a disposable glass Pasteur pipette to analyse the fatty acids. Fatty acids (FAs) were trans-esterified to form methyl esters, using 0.5 N sodium hydroxide in methanol and 14% boron trifluoride in methanol [64]. Analysis was performed using an initial isothermic period (40 °C for 2 min). After that, the temperature was increased from 4 °C/min to 230 °C. Finally, an isothermic period of 230 °C for 10 min followed. Fatty acid methyl ester (FAME) n-hexane (1 μL) was injected into the column using a Varian CP 8400 Autosampler. The injection port and detector were both maintained at 250 °C. Hydrogen, at 45 pounds per square inch (psi), functioned as the carrier gas, while nitrogen was employed as the makeup gas. Galaxy Chromatography Software (Galaxie CDS) was used to record the chromatograms.

Fatty acid methyl ester samples were identified by comparing the retention times of FAME peaks from the samples with those of standards obtained from Supelco (Supelco 37 Component Fame Mix 47885-U; Sigma-Aldrich Aston Manor, Pretoria, South Africa). All other reagents and solvents were of analytical grade and were obtained from Merck Chemicals (Pty Ltd., Halfway House, Johannesburg, South Africa). Fatty acids were expressed as a percentage of the total of all FAs present in the sample. The saturated fatty acid (SFA) percentage of the two commercialised chips was calculated as follows:% SFA = SFA (g)/totalfat (g) × 100 (1)

The following FA combinations were calculated: omega-3 (n-3) FAs, omega-6 (n-6) FAs, total SFAs, total monounsaturated fatty acids (MUFAs), polyunsaturated fatty acids (PUFAs), PUFA/SFA ratio (P/S), and n-6/n-3 ratio.

#### 2.7.4. Determination of the Mineral Content

A spectroscopic analysis of the chips’ mineral content was conducted using atomic absorption flame spectroscopy for calcium and magnesium [48,65] and flame emission spectroscopy for sodium and potassium [66]. Firstly, the calcium and magnesium contents of the samples were determined by the dry ashing AOAC method 927.02. The remaining calcium and magnesium quantities were analysed using a double-beam atomic absorption spectrometer, specifically the Analyst 300 Perkin Elmer model. A calcium standard with a concentration of 1000 parts per million (ppm) was used to generate a calibration curve. The spectrometer was operated with 12 psi of dry air and 70 psi of acetylene, using a 422.7 nanometer (nm) flame, a ten milliampere (mA) lamp current, and a 0.7 nm slit width. The average of the calcium and magnesium contents were recorded for each sample.

Secondly, flame emission spectroscopy for sodium and potassium was conducted by preparing standard sodium and potassium solutions using their respective salts. These solutions were then aspirated into a flame, atomised, and excited by the flame’s heat. The excited atoms emitted characteristic wavelengths of light, which were measured by a detector and converted into a signal. The intensity of the signal was proportional to the concentration of the element in the sample. Calibration curves were constructed using standard solutions of known concentrations, and the concentrations of sodium and potassium in the samples were determined by comparison with the calibration curves. The intensity of the light emitted by the flame was measured and used to determine the concentrations of sodium and potassium in the sample.

Lastly, phosphorus was examined using the AOAC method 931.01 [67]. This involved burning 1-g samples in porcelain crucibles for 6 h at 500 °C in a Carbolite C.E. furnace and then dissolving the ash in 100 mL volumetric flasks with distilled deionised water and a small amount of concentrated hydrochloric acid. The amount of phosphorus in the solution was determined using colorimetric measurements with a Spectronic 20 instrument (Milton Roy Co., Cambridge, UK).

### 2.8. Statistical Analysis

The sensory and nutrient data were analysed using ‘R’, R libraries—Performance Analytics for correlations, and an Excel add-on: Sensory Analysis with XLSTAT [68].

An analysis of variance (ANOVA) was utilised to examine whether there were any differences in means among the different samples. The Tukey-HSD post-hoc test was employed to identify specific significant differences between the samples, and these differences were then denoted in the tables using superscripts.

The survey data were captured on the EvaSys© V8.2 software and imported into the XLSTAT sensory analysis package for Excel (Microsoft 365) [68]. The frequencies of each question and their possible answers were calculated. Furthermore, the data was analysed descriptively using the SPSS (Statistical Package for the Social Sciences) version 28 package. A reliability analysis was performed by calculating Cronbach’s alpha for the questionnaire.

## 3. Results and Discussion

### 3.1. Demographics

The demographics of the four sensory panels and the respondents who completed the questionnaire are shown in Table 1. Most respondents were under 35 years of age, and as the study was conducted at a university, most were students. The majority of the respondents were female. Research shows that in most households, the female gender is responsible for most households’ food choices and consumption decisions [69]. The demographics for the questionnaire showed that the ages of more than half of the respondents were 18 to 24 (53.4%), followed by 25–35 (25.3%), 36–59 (19.7%), and 60 years and older (1.6%) (Table 1).

### 3.2. Sensory Analysis

#### 3.2.1. Panel 1: Vegetarian Nuggets

In the first panel to evaluate the vegetarian nuggets, the overall liking (hedonic scale) results (Table 3) showed that the maize and potato nugget samples had the highest ratings of 6.41 and 6.80, respectively (like slightly). The chickpea nugget was rated significantly lower (*p* ≤ 0.05) (5.29 = neither like nor dislike).

The chickpea nugget had the lowest JAR results (Table 4), while the maize nuggets were ranked the highest for all attributes. The JAR results (Table 4) showed that one attribute of maize nuggets was JAR (81.0% for appearance). As previously mentioned, good performance for the JAR score was established at 75% for every attribute. The mean drop values for “too much” rankings were above 1.00 for the three nugget products, suggesting reformulations to improve the attributes. The penalty for all four attributes and samples was unacceptable (higher than 1.0). It ranged from 1.2 (maize nugget sample appearance and chickpea nugget mouthfeel) to 2.0 (potato sample taste). The significant *p*-values (Table 4) for all attributes showed that reformulations were required to improve every attribute. Thus, maize nuggets had the best scores among the three samples for JAR; however, overall, the nuggets had to be reformulated for more acceptable consumer acceptance.

#### 3.2.2. Panel 2: Vegetarian Burgers

In the second panel to evaluate the vegetarian burgers (Table 3), the overall liking (hedonic scale) of the black bean sample had the highest rating (6.67 = like slightly). It was significantly higher (*p* ≤ 0.05) than that of the maize burger (5.98 = neither like nor dislike). The chickpea burgers were not significantly different from the other burger samples, as their overall liking (6.06) was also in the ‘like slightly’ category. In Table 5, the JAR results show that the appearance of the maize burger came closest to being accepted (75.0%), that is, 72.9%. The black bean sample had the highest JAR ranking for taste (57.6) and the lowest ranking for appearance (31.8). The black bean burger and chickpea burger were similar and higher than the maize burger in terms of mouthfeel. The mean drop penalties were above 1.0 for all samples except for “not enough” for black bean burger samples (0.9) and “not enough” for aroma for the maize burger sample (0.5) (Table 5). The maize burger samples also had the highest mean drop (2.5) for the “too much” taste. The penalty for all attributes, except the aroma of the maize burger (0.9), was above 1.0, meaning that reformulations for all attributes would be required. However, the maize burger sample was the most preferred overall, as it had the highest JAR scores and the lowest penalties (Table 5).

#### 3.2.3. Panel 3: Chip Samples

The overall liking results (Table 3) of the third panel for chip samples showed that the deep-fried chip had the highest rating (7.47 = like moderately) and differed significantly (*p* ≤ 0.05) from the air-fried 1 chip (5.43 = neither like nor dislike), and the air-fried 2 chip (5.48). However, the JAR results (Table 6) showed that two of the JAR scores were above the 75% benchmark (75.9 = taste and 83.5 = mouthfeel), and one was close to 75% (73.4%). However, the appearance was lower at 51.9%. None of the JAR scores for the other two air-fried chip samples were more than 50.6%. None of the deep-fried chip sample mean drops were below the benchmark of 1.0. The other two chip samples had higher mean drops. However, the mean drops for mouthfeel were closer to 1.0, showing that the panel found the mouthfeel to be more acceptable and scored higher (Table 6). The penalty was below the threshold of 1.0 for the deep-fried chip and higher for the other two chip samples. Thus, the results showed that the deep-fried sample was JAR, with no penalties, and the standardised difference was low, meaning that it was the best product test of the three panels of the nuggets, burgers, and chips (Table 6).

However, the score for appearance was not JAR, and at 52% (Table 6), it was decided to perform one final test to determine if the appearance could be improved. Appearance is one of the most significant sensory attributes of fresh and processed food products that consumers use to determine freshness and flavour quality. Thus, colour is one of the most essential attributes of food appearance and influences consumers’ choices. Colour is well correlated with other sensory quality attributes, and consumers associate pleasant flavours with attractive colours [70]. The colour of food communicates different meanings. For instance, some colours are associated with ripeness (green and red), whereas others are associated with spoilage (browning) [70]. In fact, a subconscious judgement is made about a product within 90 s, and 62–90% of consumers make this judgement based solely on the product’s colour because colour registers much faster than text or complex graphics. Additionally, nearly 85% of consumers say that colour is the most crucial factor when choosing a product [71]. Additionally, people associate specific colours with particular tastes [72]. Humans begin associating specific colours with foods from birth and equating these colours to particular tastes and flavours throughout life [73].

#### 3.2.4. Panel 4: Flavoured Chips

It was seen that colour and flavour are associated with each other. Thus, it was decided to use flavouring for the deep-fried sample as it also imparts colour to a product. Therefore, the reformulation in the fourth panel involved seasoning and colouring with two known and familiar flavourings: chutney (brown) and tomato (red).

In panel four, the overall liking results showed (Table 3) that the hedonic ratings for the chutney flavour sample (7.35) and the sweet tomato sample (7.23) were “like moderately” in the overall liking test and were not significantly different. The JAR results (Table 7) showed that the chutney sample scored above the benchmark of 75% for taste (78.8%) and above 70% for aroma (70.7%), appearance (72.7%), and mouthfeel (74.7%). The tomato chip samples received slightly lower scores for taste (68.7%) and mouthfeel (64.7%), and above 70% for aroma (71.7%) and appearance (73.3). Although the JAR scores (Table 7) showed that the flavoured samples were more acceptable, the mean drop values were generally around the benchmark of 1.0, except for the appearance of the tomato sample. This could be interpreted as more improvements could be made regarding the amount of flavouring added if the development continued. The penalty values also showed that more precise improvements to the flavours could be made to improve the product, as the standardised difference also revealed that the effect of the attributes was fairly strong and very strong (Table 7).

### 3.3. Survey

Section one (demographics) of the survey’s results was discussed in Section 2.4 (Table 1) and reported in Section 3.1. Section two of the survey aimed to provide insight into respondents’ likes and dislikes for the sensory attributes of nixtamalised maize products (Table 8). Furthermore, section two also recorded the results of the respondents’ willingness to consume, purchase, prepare, and order these products in a restaurant or recommend the products to family and friends (Table 8).

In Table 8, regarding appearance, more than half (58.3%) of the respondents liked the products, while a few (4.9%) did not like how the products looked. For the aroma of nixtamalised products, over half (54.4%) of the respondents liked it, and a small percentage (8.9%) did not like the aroma. More than one-third (38.8%) of the respondents neither liked nor disliked the taste of the products, whereas more than half (52.6%) liked the taste. A low percentage (8.5%) of respondents disliked the taste of nixtamalised products. More than half (57.8%) of the respondents liked the product texture, while 29.7% remained neutral (Table 8). The sensory aspects of new food products are essential for consumer acceptance [74].

The results in Table 9 show that most (78.5%) respondents agreed to eat the products again; 15.3% neither disagreed nor agreed with this statement, while a few (6.1%) disagreed with the repeated consumption of nixtamalised products. Most respondents (60.8%) agreed that they would purchase nixtamalised products in a grocery store; 22.0% disagreed or agreed with this statement, while 17.2% disagreed. More than half of the respondents (57.2%) agreed to prepare nixtamalised products at home for their families, while 21.7% neither disagreed nor agreed with this statement. However, 21.1% disagreed that they would prepare the products themselves. Almost half (45.5%) agreed that they would order nixtamalised maize products in a restaurant. However, 33.7% disagreed with the possibility of ordering any of the products in a restaurant, resulting in the most significant disagreement among all the statements listed. When asked if the respondents would recommend the products to family and friends, most (62.2%) agreed with this statement, 21.7% neither disagreed nor agreed, and 16.1% disagreed. Thus, most respondents agreed that they would recommend the products to family and friends.

Table 10 shows the results in response to the conditions of the consumption of nixtamalised products. In response to the statement “…I have more knowledge about the health benefits of nixtamalised maize than I do at present”, a few (15.6%) of the respondents disagreed with this statement; 19.0% neither disagreed nor agreed, and the majority (65.4%) agreed. This is supported by [75], who stated that reliable scientific-based information about novel foods could influence consumer beliefs and the acceptance of new food products. The location of food sources is a critical indicator of food accessibility for consumers [76]. The results for the statement (Table 10), “…it is available for purchase in my local supermarket” showed that a few (15.7%) respondents disagreed with this statement, 19.5% neither disagreed nor agreed, and the majority (64.8%) agreed.

For the statement, “…my friends and family use nixtamalised maize products”, a few (26.9%) of the respondents disagreed with the statement; 24.7% neither disagreed nor agreed, while almost half (48.4%) agreed (Table 10). East, Hammond, and Lomax (2008) found that positive word of mouth can positively influence a consumer’s decision to buy and use a product [77]. The results for the statement, “nixtamalised maize meal is cheaper than other maize meal or maize kernels”, showed that only a few (14.8%) disagreed, almost half (49.7%) neither disagreed nor agreed, and 35.5% agreed. For the statement, “nixtamalised maize meal is similarly priced compared to other maize meal”, the results showed that a small number (7.0%) of respondents disagreed; almost half (49.5%) neither disagreed nor agreed, while 43.5% agreed (Table 10). According to the statement, “when I do not have access to any other food”, less than a quarter (21.7%) of respondents disagreed, 27.2% neither disagreed nor agreed, and just over a half (51.1%) agreed.

Thus, most panellists liked the nixtamalised products, showed their willingness to consume, purchase, prepare, order, and recommend nixtamalised products, and were neutral or agreed to the conditions in which the products would be consumed or purchased.

### 3.4. Comparison of the Nutrient Content to Commercial Maize Chips

In the context of South Africa, where poorer communities often struggle to afford a diversified diet, the feasibility and benefits of nixtamalisation as a process of nutritional improvement remain in question. Hence, the primary objective of this discussion is to illuminate the dietary advantages and challenges linked to nixtamalisation, thereby assessing its potential to alleviate nutritional deficiencies within the broader context. By addressing the challenges surrounding affordability and dietary diversification, this exploration seeks to evaluate how nixtamalisation could serve as a viable solution for mitigating nutritional deficiencies.

The process of nixtamalisation causes essential changes in the morphology and rheological characteristics of starch, a significant component of maize. The cooking of maize grains in an alkaline solution of Ca(OH)_2_ could be crucial for human consumption of this maize since nixtamalised products, such as tamales and tortillas, are widely consumed in Mexico and by Mexican people living in the United States [78]. The relatively high temperature during cooking of the grain (between 85 °C and 100 °C) and the pH value (>12) facilitate diverse transformations of grain components. Among these are the degradation of the pericarp, the loss of soluble proteins (mainly albumin and globulin), and the partial gelatinisation of starch. During grinding, additional gelatinisation of starch is carried out, and other transformations in grain components are produced since the masa is a mixture consisting of starch polymers (amylose and amylopectin) mixed with partially gelatinised starch and intact granules, endosperm parts, and lipids. All these components form a heterogeneous and complex matrix [78].

#### 3.4.1. Comparison of the Three Samples (Table 11)

The maize chips analysed in this study (*n* = 3) were compared to the nutrient content of commercial chips; the information was sourced from an online database (Table 11) [79,80].

#### 3.4.2. Energy Content

The energy content of the commercial chips (1944 kJ/100 g and 2142 kJ/100 g, respectively) was lower than that of the nixtamalised maize chip samples (3203 kJ/100 g) (Table 11). All three products exhibit a high energy content; only food items containing 170 kJ/100 g or less are considered to have a low energy content [81] (Table 11). The high energy content of maize chips could be attributed to the dry-heat cooking method of deep-frying, in which the fried product absorbs oil [82]. During deep-frying, the product is submerged in hot oil at temperatures ranging from 120 °C to 180 °C. The heat from the hot oil is transferred to the inside of the product, while the internal moisture is released as steam. This results in rapid heat transfer and shorter cooking times [83].

Additionally, deep-frying yields distinctive sensory characteristics, such as the change of colour, crust development and gelatinisation [84]. The release of steam during frying results in the development of a porous surface that improves the crispy and crunchy texture of deep-fried potato chips [85,86]. However, it also increases the capacity to absorb oil during and after frying [44]. Therefore, deep-fried foods are high in fat and offer a higher energy (kJ) content than non-fried food products [87].

#### 3.4.3. Protein Content

Cereal grains provide human populations from developing countries with about 50% of their dietary protein [88]. More importantly, 70% of the protein intake for economically disadvantaged people comes from cereals and grains. However, cereals and grains do not provide a nutritionally balanced source of protein, such as the essential amino acids lysine and tryptophan. These amino acids are essential for human and animal health [88].

However, nixtamalisation causes changes in proteins through cross-linking, hydrophobic interaction, degradation, denaturation, or the formation of lysinoalanine, which affects the protein content [89]. Nixtamalisation causes a decrease in the protein digestibility of maize from 66 to 44.44% [90]. Variations in protein structure through the development of secondary crosslinks or iso-peptide bonds may reduce digestibility by blocking the active sites of enzyme attack or inducing the formation of compounds that inhibit digestive enzymes [33]. Although protein digestibility is reduced, nixtamalisation has been reported to improve the availability of the essential amino acids lysine and tryptophan in cooked tortillas, thus improving overall protein quality [31]. Lysine and tryptophan increased by 2.8 times; additionally, the ratio of isoleucine to leucine increased by 7.8 times [30]. Traditional nixtamalisation causes a significant increase in lysine and tryptophan contents [91].

The protein content of the nixtamalised maize chips (6.64 g/100 g) compared well with the commercial maize chips (Table 11) and could be compared with the protein content in maize [92]. Maize chips are widely recognised for having low protein content [93].

#### 3.4.4. Fat Content

The nixtamalised maize chips had a total fat content of 23.72 g/100 g (Table 11), which was lower compared to that of commercial chip sample 1 (31.00 g/100 g) but higher than that of commercial chip sample 2 (20.70 g/100 g), as shown in Table 11. The total fat (FAs) content of maize decreases as the kernel is nixtamalised; this results from the loss of the seed coat, tip cap, and possibly part of the germ [94]. The nixtamalised maize chips had a total fat content of 23.72 g/100 g, which was lower than commercial maize chip 2 (31.00 g/100 g) but higher than commercial maize chip 1 (20.70 g/100 g) (Table 12).

The total fat (or FAs) content (Table 11) in the nixtamalised maize chips was attributed to deep-frying of the product in oil. One of the main concerns with deep-fried foods is their high total fat content, which contributes to obesity, high cholesterol levels, or high blood pressure [95,96]. There are mitigating viable strategies to reduce oil uptake in deep-fried foods. These strategies include modifying the surface of the food by using a batter and breading, modifying the frying oil or technique, vacuum frying, microwave heating, de-oiling, and air frying [97]. However, air frying could be the best method for reducing oil uptake in potato chips [98].

The SFA content of nixtamalised maize chips was considerably lower compared to that of the commercial chip samples, with 40.1% and 39.4% FAs, respectively (Table 11). MUFA and PUFA contents in Doritos and Fritos were not stipulated in the nutrition label of the commercialised chips.

#### 3.4.5. Sodium

South Africa has developed mandatory legislation to reduce salt content in processed foods [99]. The sodium legislation was passed in 2013 by the South African Department of Health, which restricts the maximum levels of salt permitted in commonly consumed foods [100]. ‘’Ready-to-eat savoury snacks” such as extruded, expanded, or puffed snacks made from maize, potato, rice, and other cereals should contain no more than 800 mg/100 g of salt [101]. Thus, the sodium content of nixtamalised maize and commercial chips fell below the regulatory concentrations (Table 11). Maize naturally has a very low sodium content, typically around 35 mg/100 g. However, many maize food products acquire sodium during processing. Salt is often added for flavour enhancement purposes, significantly increasing sodium content, especially in snack foods [102].

### 3.5. Nutrient Content of the Nixtamalised Maize Chip Sample

The nutrient content of the chutney-flavoured nixtamalised maize chip per 100 g (*n* = 3) is given in Table 12. Common maize varieties contain about 12% insoluble fibre, with processed corn having a reduced amount [33]. Nixtamalised maize commonly has a 10 to 12% decrease in insoluble fibres due to the pericarp removal [33]. The fibre content analysis showed that the NDF (15.87 g/100 g) in the nixtamalised maize chips was higher than the ADF (1.32 g/100 g) (Table 12). This could be compared to the insoluble fibre content in popcorn (14 g/100 g), which is considered a high-fibre snack.

Nixtamalised maize serves as a source of calcium due to the cooking and steeping of maize in a Ca(OH)_2_ solution [102,103]. Interestingly, non-nixtamalised whole dry maize kernels are very low in calcium, containing only 7 mg/100 g [102]. Therefore, maize itself is not a reliable source of calcium. The average calcium content of nixtamalised maize was 170 mg/100 g, an almost 18-fold increase [30]. The calcium content of the nixtamalised maize chip (163.33 mg/100 g) (Table 12) was slightly higher than that of milk (123 mg/100 g) [44]. Thus, nixtamalised maize chips could be considered an excellent dietary calcium source.

Maize contains about 127 mg of magnesium per 100 g of whole kernels (Table 12). It has been established that magnesium forms complexes with phytates that hinder its absorption [104]. However, magnesium content increased significantly after nixtamalisation and subsequent preparation into dough and tortillas, from 165 mg/100 g in unprocessed maize to 180 mg/100 g in dough and tortillas [105]. The average magnesium content for the nixtamalised maize chips was lower (53.67 mg/100 g) (Table 12).

Phosphorus is a vital mineral in the human body, which is involved in many essential functions such as the cell energy cycle, regulating the body’s acid-base balance, and mineralising bones and teeth. The phosphorus content in the nixtamalised maize chips was measured at 566 mg/100 g (Table 12), which is comparable to that in protein-rich foods like dairy, meat, and fish, which are known to be primary sources of phosphorus [106].

However, the phosphorus content in maize is mainly phytic acid [102]. Phytic acid, found in cereals, legumes, oil seeds, and nuts, is the primary storage form of phosphorus, constituting approximately 1 to 5% of the total weight [107]. Approximately 90% of phytic acid is concentrated in the germ of the maize kernel, with the remaining 10% present in the aleurone layer. Phytic acid reduces the bioavailability of other nutrients [108]. The physical removal of maize bran and germ has been found to reduce phytic acid content by approximately one-quarter to one-third [109]. Various heat treatments applied to fresh and dried maize have been shown to decrease the phytic acid content. Nixtamalisation reduces phytic acid by about 20% [30]. 

Whole-grain maize flour contains relatively high potassium levels, with a content of approximately 315 mg/100 g [102]. On the other hand, degermed flour, which has had the germ removed, exhibits a substantially lower potassium content, measuring around 90 mg/100 g. This indicates that the potassium content can vary depending on the processing and removal of specific components from the maize flour. Further-processed maize products, such as tortillas and extruded chips, contain 186 and 144 mg/100 g, respectively [102].

Saturated fatty acids (SFAs) are positively associated with cardiovascular diseases and high cholesterol levels. It has been proven that replacing or substituting SFAs with polyunsaturated fats (PUFAs) and monounsaturated fats (MUFAs) reduces the risk of cardiovascular disease and other health conditions related to SFAs [110]. As shown in Table 12, the highest percentage of FAs (23.72, as indicated in Table 11) in the nixtamalised maize chips was PUFAs (49.0% of FAs), while 40.0% of FAs were MUFAs (Table 12). The SFAs made up the least (11.0%) of the FAs in the nixtamalised maize chips, which was considerably lower than that of the commercial chips. Olive and sunflower oils have been used in the formulation of nixtamalised maize chips, and these oils are the major sources of PUFAs and MUFAs [44].

The omega-3 content in the nixtamalised maize chips was limited to 0.17% of FAs (Table 12). Eicosapentaenoic acid is a polyunsaturated omega-3 fatty acid, in which the first double bond is three carbons from the methyl (CH3) end, and it plays a vital role in both human and animal health [44,111]. This fatty acid is commonly found in seafood, such as salmon and herring.

Omega-6 fatty acids account for almost half (48.41%) of the FAs present in the nixtamalised chips (Table 12). The lipids available in nixtamalised maize are richer in linoleic groups than in oleic groups [112]. Linoleic acid (C18:2c9,12 (n-6)) is an essential omega-6 PUFA that cannot be synthesised by the body, requiring it to be consumed through the diet. It is primarily found in vegetable oils, nuts, seeds, meats, and eggs [113]. Polyunsaturated fatty acids play vital roles in many biological processes in the body, such as being endogenous mediators for cell signalling and being involved in regulating gene expression [114].

The findings indicated that the omega-6 content was much higher than the omega-3 content (Table 12), likely due to the use of sunflower oil in cooking, which contains 69.0% linoleic acid [115]. Linoleic acid and oleic acid are the primary FAs present in maize, accounting for over 98.0% of the FA content in high-oil maize hybrids [116].

## 4. Conclusions

An acceptable, nutritious, and consumer-acceptable nixtamalised maize-based product was developed by employing consumer-led methods that could be produced in a home-based environment. It received favourable sensory evaluations, and panel members were willing to consume, purchase, and prepare them.

A number of limitations have to be considered. The products were unique, and the formulations could not be published; the panels comprised a limited number of consumers (less than 100) due to limited space. The panel members were screened for being regular users of maize products and pre-trained to perform the tasks satisfactorily. However, other populations may respond differently, and the results cannot be widely applied.

Although the current study shows encouraging results, the researcher recommends that further research on a larger scale is warranted. Further studies are required to fully elucidate the potential impact of nixtamalisation on the nutritional value of maize products and to assess the feasibility of entrepreneurial action and scaling up nixtamalisation processes for commercial production.

Undoubtedly, the potential impact of this study on the future prospects of maize and maize products in South Africa cannot be overstated, particularly with respect to addressing food security and malnutrition among low-income populations. Given these factors, the current research presents a significant milestone in achieving the full potential of nixtamalisation as a means of adding value to maize-based products in South Africa.

## Figures and Tables

**Table 1 foods-13-02896-t001:** Age and gender of the participants in the four sensory panels and the questionnaire.

		Nuggets		Burgers		Original Chips		Flavoured Chips		Questionnaire	
Age		*n*	%	*n*	%	*n*	%	*n*	%	*n*	%
	≤35	57.0	72.2	64.0	77.1	59.0	79.7	71.0	71.7	140.0	75.3
	>35	22.0	27.8	19.0	22.9	15.0	20.3	28.0	28.3	38.0	20.4
	Not disclosed									8.0	4.3
	Total	79.0	100.0	83.0	100.0	74.0	100.0	99.0	100.0	178.0	95.7
Gender											
	Male	14.0	17.7	14.0	16.9	13.0	17.6	14.0	14.1	32.0	17.2
	Female	65.0	82.3	69.0	83.1	61.0	82.4	83.0	83.8	145.0	78.0
	Other							1.0	1.0	1.0	0.5
	Not disclosed							1.0	1.0	8.0	4.3
	Total	79.0	100.0	83.0	100.0	74.0	100.0	99.0	100.0	186.0	100.0

**Table 2 foods-13-02896-t002:** Evaluation criteria for the four sensory attributes of each product (vegetarian nuggets, vegetarian burgers, and original and flavoured chips) used to rate the samples on a five-point JAR scale.

		Vegetarian Nuggets	Vegetarian Burgers	Original and Flavoured Maize Chips
Aroma		Spicy	Spicy	Oily/corn/starch
	5	Much too spicy	Much too spicy	Much too strong
	4	Somewhat too spicy	Somewhat too spicy	Somewhat too strong
	3	Just-about-right	Just-about-right	Just-about-right
	2	Somewhat too bland	Somewhat too bland	Somewhat too low
	1	Much too bland	Much too bland	Much too low
Appearance		Colour	Colour	Colour
	5	Much too brown	Much too brown	Much too brown
	4	Somewhat too brown	Somewhat too brown	Somewhat too brown
	3	Just-about-right	Just-about-right	Just-about-right
	2	Somewhat dull	Somewhat dull	Somewhat dull
	1	Much too dull	Much too dull	Much too dull
Taste		Savoury	Savoury	Savoury
	5	Much too savoury	Much too savoury	Much too savoury
	4	Somewhat too savoury	Somewhat too savoury	Somewhat too savoury
	3	Just-about-right	Just-about-right	Just-about-right
	2	Somewhat too bland	Somewhat too bland	Somewhat too bland
	1	Much too bland	Much too bland	Much too bland
Mouthfeel		Crunchy	Soft	Crispy
	5	Much too crunchy	Much too soft	Much too crispy
	4	Somewhat too crunchy	Somewhat too soft	Somewhat too crispy
	3	Just-about-right	Just-about-right	Just-about-right
	2	Somewhat too soft	Somewhat too hard	Somewhat too soft
	1	Much too soft	Much too hard	Much too soft

**Table 3 foods-13-02896-t003:** Overall liking of products ranked on a 9-point hedonic scale.

Product	Sample	Overall Liking Hedonic Scaling ± SD
Nuggets	Maize	6.80 ^b^ ± 1.57
(*n* = 79)	Potato	6.41 ^b^ ± 1.7
	Chickpea	5.29 ^a^ ± 1.83
Burgers	Maize	5.98 ^c^ ± 1.93
(*n* = 83)	Black bean	6.67 ^d^ ± 1.40
	Chickpea	6.06 ^cd^ ± 1.74
Original chips	Deep-fried	7.47 ^e^ ± 1.06
(*n* = 74)	Air-fried 1	5.43 ^f^ ± 1.77
	Air-fried 2	5.48 ^f^ ± 1.69
Flavoured chips	Chutney	7.35 ^g^ ± 1.37
(*n* = 99)	Sweet tomato	7.23 ^g^ ± 1.37

Different superscripts indicate significant differences for each product (*p* < 0.05).

**Table 4 foods-13-02896-t004:** JAR%, mean drop, and penalty analysis for maize, potato, and chickpea nugget samples for panel one.

		Aroma			Appearance			Taste			Mouthfeel		
		Not Enough	JAR	Too High	Not Enough	JAR	Too High	Not Enough	JAR	Too High	Not Enough	JAR	Too High
Maize nugget	JAR %	22.8	65.8	11.4	15.2	81.0	3.8	20.2	64.6	15.2	40.0	54.4	7.6
	Mean drop	1.2 *		1.7	1.0		1.7 *	1.9		1.6	1.1 *		2.4
	Penalty	1.4			1.2			1.7			1.3		
	*p*-value	0.00			0.01			<0.00			<0.00		
Potato nugget	JAR %	6.3	58.2	35.5	3.8	58.2	38.0	5.1	62	32.9	6.3	49.4	44.3
	Mean drop	0		1.6 *	0		1.5 *	0.7		2.2 *	1.2		1.6 *
	Penalty	1.3			1.3			2.0			1.6		
	*p*-value	0.00			0.00			<0.00			<0.00		
Chickpea nugget	JAR %	5.1	32.9	62.0	17.7	36.7	45.6	12.7	31.6	55.7	22.8	51.9	25.3
	Mean drop	1.0		1.9 *	0.3		1.7 *	1.3		2.0 *	1.0		1.4 *
	Penalty	1.8			1.3			1.9			1.2		
	*p*-value	<0.00			0.00			<0.00			<0.00		

* Indicates significant values (*p* < 0.05).

**Table 5 foods-13-02896-t005:** JAR%, mean drop, and penalty analysis for maize, black bean, and chickpea burger samples for panel two.

		Aroma			Appearance			Taste			Mouthfeel		
		Not Enough	JAR	Too High	Not Enough	JAR	Too High	Not Enough	JAR	Too High	Not Enough	JAR	Too High
Maize burger	JAR %	9.4	56.5	34.1	1.2	72.9	25.9	20.0	50.6	29.4	1.2	42.4	56.5
	Mean drop	0.5		1.0 *	0		1.4 *	1.6 *		2.5 *	2.9		1.6 *
	Penalty	0.9			1.3			2.1			1.9		
	*p*-value	0.03			0.01			<0.00			<0.00		
Black bean burger	JAR %	30.6	58.8	10.6	64.7	31.8	3.5	20.0	57.6	22.4	28.3	52.9	18.8
	Mean drop	1.3 *		1.6	0.9 *		1.7	1.7 *		1.4 *	1.2 *		1.7
	Penalty	1.3			1.0			1.5			1.4		
	*p*-value	<0.00			0.00			<0.00			0.00		
Chickpea burger	JAR %	5.9	50.6	43.5	36.5	54.1	9.4	5.9	35.3	58.8	27.1	52.9	20.0
	Mean drop	2.5		1.2 *	1.0 *		2.1	1.8		1.7 *	2.1 *		1.6 *
	Penalty	1.4			1.3			1.7			1.9		
	*p*-value	0.00			0.00			<0.00			<0.00		

* Indicates significant values (*p* < 0.05).

**Table 6 foods-13-02896-t006:** JAR%, mean drop, and penalty analysis of deep-fried, air-fried 1, and air-fried 2 chip samples for panel three.

		Aroma			Appearance			Taste			Mouthfeel		
		Not Enough	JAR	Too High	Not Enough	JAR	Too High	Not Enough	JAR	Too High	Not Enough	JAR	Too High
Deep-fried	JAR %	21.5	73.4	5.1	29.1	51.9	19.0	5.1	75.9	19.0	15.2	83.5	1.3
	Mean drop	0.3		0.6	0.1		0.0	0.0		0.7	0.5		0.5
	Penalty	0.3			0.0			0.5			0.4		
	*p*-value	0.3			0.9			0.05			0.2		
Air-fried 1	JAR %	21.5	40.5	38.0	2.5	35.4	62.0	12.7	27.9	59.5	24.0	49.4	26.6
	Mean drop	1.9 *		1.8 *	2.6		1.8 *	1.9		2.4 *	1.05		1.00
	Penalty	1.8			1.8				2.3		1.0		
	*p*-value	<0.00			<0.00				<0.00		0.01		
Air-fried 2	JAR %	21.5	39.3	39.2	1.3	50.6	48.1	8.9	29.1	62.0	20.3	30.4	49.3
	Mean drop	1.2 *		0.7	3.0		0.9 *	1.8		1.6 *	0.9		1.3 *
	Penalty	0.9			1.0			1.7			1.2		
	*p*-value	0.02			0.01			<0.00			0.00		

* Indicates significant values (*p* < 0.05).

**Table 7 foods-13-02896-t007:** JAR%, mean drop, and penalty analysis chutney and tomato flavour chip samples for panel four.

		Aroma			Appearance			Taste			Mouthfeel		
		Not Enough	JAR	Too High	Not Enough	JAR	Too High	Not Enough	JAR	Too High	Not Enough	JAR	Too High
Chutney flavour chips	JAR %	14.1	70.7	15.2	26.3	72.7	1.0	12.1	78.8	9.1	16.2	74.7	9.1
	Mean drop	1.1		1.1	1.6 *		1.8	1.5		1.5	1.5		0.7
	Penalty	1.1			1.6			1.5			1.2		
	*p*-value	0.00			<0.00			<0.00			0.00		
Tomato flavour chips	JAR %	5.1	73.7	21.2	1.0	71.7	27.3	26.3	68.7	5.0	29.3	64.7	6.0
	Mean drop	0.4		0.6	1.5		0.8 *	1.6 *		1.7	1.6 *		1.6
	Penalty	0.8			0.6			1.6			1.6		
	*p*-value	0.00			0.07			<0.00			<0.00		

* Indicates significant values (*p* < 0.05).

**Table 8 foods-13-02896-t008:** Likes and dislikes for the sensory attributes of nixtamalised products.

Question		Do not Like at All	Do not Like	Neither Like or Dislike	Like	Like it Very Much	Missing Values	Total
I liked the appearance of the product	*n*	1	8	67	68	38	4	186
	%	0.5	4.4	36.8	37.4	20.9		100
I liked the aroma of the product	*n*	1	15	66	69	29	6	186
	%	0.6	8.3	36.7	38.3	16.1		100
I liked how the product tasted	*n*	1	14	68	61	31	11	186
	%	0.6	8.0	38.9	34.9	17.7		100
I liked the texture of the product	*n*	0	22	53	72	31	8	186
	%	0.0	12.4	29.8	40.4	17.4		100

**Table 9 foods-13-02896-t009:** Willingness to consume, purchase, prepare, order, and recommend nixtamalised products.

Question		Strongly Disagree	Disagree	Neither Agree or Disagree	Agree	Strongly Agree	Missing Values	Total
I will eat this food product again	*n*	1	10	28	96	47	4	186
	%	0.5	5.5	15.4	52.7	25.8		100
I will purchase this product in a grocery store	*n*	3	28	40	69	41	5	186
	%	1.7	15.5	22.1	38.1	22.7		100
I will prepare this product for my family	*n*	4	34	39	69	34	6	186
	%	2.2	18.9	21.7	38.3	18.9		100
I will order this product in a restaurant	*n*	12	48	37	53	28	8	186
	%	6.7	27.0	20.8	29.8	15.7		100
I will recommend this product to my friends and family	*n*	4	25	39	77	35	6	186
	%	2.2	13.9	21.7	42.8	19.4		100

**Table 10 foods-13-02896-t010:** Conditions for consuming nixtamalised products.

Statement I Would Eat Nixtamalised Products if……		Strongly Disagree	Disagree	Neither Agree or Disagree	Agree	Strongly Agree	Missing Values	Total
…these products were given to me for free	*n*	13	43	53	48	28	1	186
	%	7.0	23.2	28.6	25.9	15.1		100
…I had more knowledge about the health benefits of nixtamalised maize than I do at present	*n*	5	24	35	82	39	1	186
	%	2.7	13.0	18.9	44.3	21.1		100
…they were available for purchase in my local supermarket	*n*	9	20	36	90	30	1	186
	%	4.9	10.8	19.5	48.6	16.2		100
…my friends and family use nixtamalised maize products	*n*	14	35	45	65	23	4	186
	%	7.7	19.2	24.7	35.7	12.6		100
…nixtamalised maize meal is cheaper than other maize meal or maize kernels	*n*	2	25	91	47	18	3	186
	%	1.1	13.7	49.7	25.7	9.8		100
…nixtamalised maize meal is similarly priced compared to other maize meal	*n*	2	11	91	65	15	2	186
	%	1.1	6.0	49.5	35.3	8.2		100
…I did not have access to any other food	*n*	8	32	50	62	32	2	186
	%	4.3	17.4	27.2	33.7	17.4		100

**Table 11 foods-13-02896-t011:** Comparison of the nutrient content as indicated on the label of two commercial maize-based chips and the chutney-flavoured nixtamalised chip samples.

Nutrient	Nixtamalised Maize Chips (*n* = 3)	Commercial Maize Chip 1	Commercial Maize Chip 2
Energy (kJ/100 g)	2303	1944	2142
Crude Protein (g/100 g)	6.64	6.70	6.20
Total Fats (FA’s) (g/100 g)	23.72	20.70	31.00
Saturated fat (% of total fats)	11.47	8.3	12.2
Sodium (mg/100 g)	706.67	634.00	685.00

**Table 12 foods-13-02896-t012:** Fibre, mineral, and fat content of the nixtamalised maize chip per 100 g (*n* = 3).

	Nutrient	
Fibre	Neutral detergent fibre (g/100 g)	15.87
	Acid detergent fibre (g/100 g)	1.32
Minerals	Calcium (mg/100 g)	163.33
	Magnesium (mg/100 g)	53.67
	Sodium (mg/100 g)	706.67
	Potassium (mg/100 g)	154.33
	Phosphorus (mg/100 g)	566.00
Fats	Total fat content (%)	23.7
Fatty acids ratios (% of total fats)	Saturated fatty acids (SFA)	11.47
	Monounsaturated fatty acids (MUFA)	39.93
	Polyunsaturated fatty acids ratio (PUFA)	48.58
	Omega-6 fatty acids	48.41
	Omega-3 fatty acids	0.17
	PUFA:SFA	4.23

## Data Availability

The original contributions presented in the study are included in the article, further inquiries can be directed to the corresponding author.

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
