# Peer review of "Developing an Acceptable Nixtamalised Maize Product for South African Consumers: Sensory, Survey and Nutrient Analysis"

_foods, 2024, doi:10.3390/foods13182896_

Round 1
Reviewer 1 Report
Comments and Suggestions for Authors
This study aimed to evaluate the nixtamalized products using overall acceptability (9-point hedonic scale), Just-About-Right (JAR) and penalty analysis. Moreover, the study took the nutrient analysis of the chutney chips. However, there are some weaknesses that need to be addressed.
1. The format of some cited references miss the authors in the manuscript, such as Page 2, reference numbered 16; Page 4, reference numbered 44; Page 6, reference numbered 57; Page 8, reference numbered 69; Page 16, reference numbered 18.
2. 2 Materials and methods
This section only contained subtitles “2.1 sample preparation” and “2.2 Statistical analysis”. Suggest to add titles according to each section of the content.
3. Page 9: “It ranged from 1.2 (maize nugget sample appearance and chickpea nugget mouthfeel) to 2.0 (chickpea sample taste).”
The penalty of potato nugget sample taste was 2.0, not chickpea sample taste.
4. Page 10: “The penalty for all the attributes except the aroma of the maize burger (0.9) was above 0.1, meaning that reformulations for all the attributes would be required”
The sentence is confused, the aroma of the maize burger (0.9) was also above 0.1.
5. Page 10-11: “The penalty was below the threshold of 0.1 for the deep-fried chip and higher for the other two chip samples”.
This sentence is confused. The penalty of aroma was 0.3 for the deep-fried chip, the penalties of taste and mouthfeel were 0.5 and 0.4, respectively. The results were not below the threshold of 0.1.
6. the first line of Page 12: continoued or continued?
7.Table 7, Table 8 and Table 9: suggest to express the data of % as point, such as 0.5, 4.4.
8. Page 5: “the panellists were expected to complete the JAR scales for four attributes, namely aroma, appearance, taste, and mouthfeel”.
The JAR analysis was explored in different products, such as nuggets, burgers, original chips and flavoured chips. Do you provide the reference materials for the four attributes? The aroma appearance, taste, and mouthfeel characteristics of the four different products were different. What is the evaluation criterion for the four attributes in different products?
Author Response
This study aimed to evaluate the nixtamalized products using overall acceptability (9-point hedonic scale), Just-About-Right (JAR) and penalty analysis. Moreover, the study took the nutrient analysis of the chutney chips. However, there are some weaknesses that need to be addressed.
1. The format of some cited references miss the authors in the manuscript, such as
|
|
Page 2, reference numbered 16; Page 4, reference numbered 44; Page 6, reference numbered 57; Page 8, reference numbered 69; Page 16, reference numbered 18. |
The citations were corrected. All citations were checked and carefully corrected. |
2. 2 Materials and methods
This section only contained subtitles “2.1 sample preparation” and “2.2 Statistical analysis”. Suggest to add titles according to each section of the content. |
Headings were added to the materials and methods section to improve the readability and to organize the content. |
3. Page 9: “It ranged from 1.2 (maize nugget sample appearance and chickpea nugget mouthfeel) to 2.0 (chickpea sample taste).” The penalty of potato nugget sample taste was 2.0, not chickpea sample taste. |
Thank you for pointing out the mistake. The sentence was corrected. |
4. Page 10: “The penalty for all the attributes except the aroma of the maize burger (0.9) was above 0.1, meaning that reformulations for all the attributes would be required” The sentence is confused, the aroma of the maize burger (0.9) was also above 0.1. |
Thank you for catching this mistake, the threshold is 1.0, therefore the sentence should read: “The penalty for all the attributes except the aroma of the maize burger (0.9) was above 1.0, meaning that reformulations for all the attributes would be required.” |
5. Page 10-11: “The penalty was below the threshold of 0.1 for the deep-fried chip and higher for the other two chip samples”.
This sentence is confused. The penalty of aroma was 0.3 for the deep-fried chip, the penalties of taste and mouthfeel were 0.5 and 0.4, respectively. The results were not below the threshold of 0.1. |
Thank you so much for spotting this mistake. The threshold should be 1.0 and not 0.1. It has been changed throughout. |
6. the first line of Page 12: continoued or continued? |
Thank you for pointing out the mistake. The sentence was corrected. |
7.Table 7, Table 8 and Table 9: suggest to express the data of % as point, such as 0.5, 4.4. |
Thank you for pointing that out. All the commas were replaced with points. |
8. Page 5: “the panellists were expected to complete the JAR scales for four attributes, namely aroma, appearance, taste, and mouthfeel”. The JAR analysis was explored in different products, such as nuggets, burgers, original chips and flavoured chips. Do you provide the reference materials for the four attributes? The aroma appearance, taste, and mouthfeel characteristics of the four different products were different. What is the evaluation criterion for the four attributes in different products? |
Apologies for the omission. The citation was added. |
Reviewer 2 Report
Comments and Suggestions for Authors
The article submitted for review concerns the developing an acceptable nixtamalized maize product for South African consumers. The manuscript submitted for review is a full article on the development a consumer-acceptable nixtamalized maize product by employing consumerled methods that could be produced in a household kitchen from whole dried maize kernels. The article provides interesting information and it is written quite well, but there are some points that could improve it. Below are some comments and suggestions to the Authors:
1. More specific terms should be added to the keywords, relating primarily to the type of products tested.
2. The article is very long, which makes it difficult to read. In many places, especially when describing methods or results, fragments are too long and do not raise important issues. Some fragments should be shortened slightly and written in a more concise and substantive/scientific language.
3. For example, in the “Introduction” section, the content regarding the historical outline (Lines 53 to 59) is unnecessary in this article. The text contained in lines 60 to 125 should be shortened, leaving the information that is key to the article. Those that correspond to the title and purpose of the research presented in this article.
4. The chapter "Materials and methods" also requires shortening. Their description is too long-winded, and it also contains information that should rather be included in the chapter on the description of the results and their discussion, for example - the text contained in lines 152 to 179. The texts in lines 179 to 181 are unnecessary, because in further description of product preparation, the equipment is replaced again. Additionally, it would be worth including in this part a graphic diagram of the entire experiment, starting from the processing of raw materials, through the preparation of various products and ending with tests. It would be more readable and understandable to a potential reader.
5. The description of sensory methods is too long. Methods should be described in a concise and specific manner. At this point, they span 2 pages of this manuscript. They need to be shortened and improved. Additionally, Table 1 and its description and the description of the survey method and how to complete it (Lines 201 to 291) are also excessively lengthy. Also, if shortened, this information would rather be included in the "Results and Discussion" section.
6. Additionally, in the "Materials and methods" section, individual sections should be separated and titled, i.e. sensory methods and other physicochemical methods, etc. In particular, the crude protein determination, the acid-detergent fibre (ADF), the neural detergent fibre (NDF), total lipids, fatty acid methyl ester, mineral content (Lines from 292 to 374), because it is difficult to find specific information regarding analyses.
7. Generally, the weakness of the work is the superficial discussion of the results obtained. Yes, they are described, but there are few reports of other, current results in the literature. Considering that the Authors obtained interesting research results, it would be worth expanding their discussion a bit and updating the references to the literature.
8. The "Conclusions" section is too long and contains too general information. It should include key quantitative results. The novelty and usefulness of the obtained research results should also be emphasized.
Author Response
The article submitted for review concerns the developing an acceptable nixtamalized maize product for South African consumers. The manuscript submitted for review is a full article on the development a consumer-acceptable nixtamalized maize product by employing consumer-led methods that could be produced in a household kitchen from whole dried maize kernels. The article provides interesting information and it is written quite well, but there are some points that could improve it. Below are some comments and suggestions to the Authors:
1. More specific terms should be added to the keywords, relating primarily to the type of products tested. |
Thank you for the suggestion. Vegetarian nuggets and burgers and nixtamalized chips were added to the keywords. |
2. The article is very long, which makes it difficult to read. In many places, especially when describing methods or results, fragments are too long and do not raise important issues. Some fragments should be shortened slightly and written in a more concise and substantive/scientific language. |
The length of the article was our primary concern from the start. It was important that the Materials and Methods include enough information to make the discussions understandable especially since JAR is not universally well-known. |
3. For example, in the “Introduction” section, the content regarding the historical outline (Lines 53 to 59) is unnecessary in this article.
The text contained in lines 60 to 125 should be shortened, leaving the information that is key to the article. Those that correspond to the title and purpose of the research presented in this article. |
The content regarding the historical outline was added with the intent to give context. Context is essential to grasp the importance and purpose of this research. In fact, reviewer 3 asked for more context in terms of the country and region. |
4. The chapter "Materials and methods" also requires shortening. Their description is too long-winded, and it also contains information that should rather be included in the chapter on the description of the results and their discussion, for example |
We are very conscious of the fact that the article is long. In fact, extending the MM to be such a large part of the article was not an easy decision. However, it was done because this type of research is not seen very often in scientific journals. It is imperative that this type of research get exposure, thus future researchers should be able to replicate the results and understand the process. This journal was specifically selected because it allows more detailed descriptions of the methods. |
- the text contained in lines 152 to 179. |
This section gives background and context to the specific products that were developed. This information was originally left out, however it was imperative to add it, questions were raised as to what the products were and how they were prepared as it had a significant influence on the product quality and nutritional content. |
The texts in lines 179 to 181 are unnecessary, because in further description of product preparation, the equipment is replaced again. |
Agreed, the sentence was removed. |
Additionally, it would be worth including in this part a graphic diagram of the entire experiment, starting from the processing of raw materials, through the preparation of various products and ending with tests. It would be more readable and understandable to a potential reader. |
The graphical abstract that was submitted with the article gives the information mentioned and the authors agree with reviewer 2 that the flow-diagram of the process gives more context to the reader. |
5. The description of sensory methods is too long. Methods should be described in a concise and specific manner. At this point, they span 2 pages of this manuscript. They need to be shortened and improved.
|
The length of the manuscript and in particular the descriptions of the sensory methods were issues that the authors discussed at length. However, a full and more precise description was included because this is the only way that the JAR method could be followed in the results. We also believe that future researchers will benefit from the fully described methods. The length of the materials and methods (MM) the reason why we chose this journal, as it does not have restrictions on the length of the manuscript or the MM. |
Additionally, Table 1 and its description and the description of the survey method and how to complete it (Lines 201 to 291) are also excessively lengthy. Also, if shortened, this information would rather be included in the "Results and Discussion" section. |
The information given in this section is pertaining to the questionnaire and would be essential when dealing with human participants. |
6. Additionally, in the "Materials and methods" section, individual sections should be separated and titled, i.e. sensory methods and other physicochemical methods, etc. In particular, the crude protein determination, the acid-detergent fibre (ADF), the neural detergent fibre (NDF), total lipids, fatty acid methyl ester, mineral content (Lines from 292 to 374), because it is difficult to find specific information regarding analyses. |
Headings were included to highlight the specific analysis. The headings should improve the readability and organize the content. |
7. Generally, the weakness of the work is the superficial discussion of the results obtained. Yes, they are described, but there are few reports of other, current results in the literature. Considering that the Authors obtained interesting research results, it would be worth expanding their discussion a bit and updating the references to the literature. |
The results were discussed in as much detail as possible. We consulted many other references and read widely on the subject but work in this field is scarce and specifically on the JAR methods are few. We hope that publishing our results will inspire more researchers to do similar studies that could expand the field. However, if we missed any relevant work, we would include it with pleasure. |
8. The "Conclusions" section is too long and contains too general information. It should include key quantitative results. The novelty and usefulness of the obtained research results should also be emphasized. |
The reviewer’s comment encouraged us to evaluate the conclusion. It was shortened and the general information was removed. The novelty and usefulness of the results were empathized. More specific directions would be appreciated in this regard. |
Reviewer 3 Report
Comments and Suggestions for Authors
The article “Developing an Acceptable Nixtamalized Maize Product for South African Consumers: Sensory, Survey and Nutrient Analysis” has been reviewed, obtaining the following comments.
In the introduction, rather than giving a complete definition of what food security is, it is also important to put into context the region or country that they want to benefit from their food proposal, in terms of moderate or severe food security (percentages).
It is necessary to give a better order to the introduction. There are parts that are mentioned and two paragraphs later they are mentioned again, as in the case of food security. It is necessary to cover topic by topic in an orderly manner, avoiding mixing them.
It is also important to put into context the consumption of corn or nixtamalized corn, for example, the basis of the Mexican diet is corn, a global comparison is important.
• It is important to divide the methodology into several clear and defined subsections
• The nixtamalization process
• The product elaboration process
• The nutritional analysis process
• The hedonic evaluation process
• And the statistical analysis
It is recommended to compare the results with previous studies, especially regarding the nutrient content of the flour obtained.
An important part of nixtamalization is the use of energy to carry it out, this is also important to consider when talking about expanding this type of food. The question is: where will the users get the energy from? Do they have money for this purpose? Is it sustainable?
It is recommended to make a more specific conclusion on the evaluated aspects, without the need to fall into making a summary of the results obtained speaking quantitatively.
Otherwise, I consider it to be an interesting and very complete work, which should address the previous recommendations with the objective that the reader can take advantage of the information in an easy and clear way.
Author Response
Reviewers comment: The article “Developing an Acceptable Nixtamalized Maize Product for South African Consumers: Sensory, Survey and Nutrient Analysis” has been reviewed, obtaining the following comments.
In the introduction, rather than giving a complete definition of what food security is, it is also important to put into context the region or country that they want to benefit from their food proposal, in terms of moderate or severe food security (percentages). |
Thank you for the suggestion, however the reason why the food security percentages were not given, was because according to percentages, South Africans are food secure. Thus, the introduction proceeds to describe why that is not the case and an argument is structured in the introduction to prove our point of view. |
It is necessary to give a better order to the introduction. There are parts that are mentioned and two paragraphs later they are mentioned again, as in the case of food security. It is necessary to cover topic by topic in an orderly manner, avoiding mixing them. |
The introduction was structured with the intent to build an argument and not only to give information. The reason for bringing up food security in more than one instance is to link food security to create context and give a holistic view of the current situation. |
It is also important to put into context the consumption of corn or nixtamalized corn, for example, the basis of the Mexican diet is corn, a global comparison is important. |
Nixtamalization is a process that is widely applied in Mexico and by the local community in Mexico. That is elucidated on in line 125. |
• It is important to divide the methodology into several clear and defined subsections • The nixtamalization process
• The product elaboration process
• The nutritional analysis process
• The hedonic evaluation process
• And the statistical analysis |
Thank you for pointing this out. Headings were added to organize the content. |
It is recommended to compare the results with previous studies, especially regarding the nutrient content of the flour obtained. |
Thank you for the suggestion. However, studies on the nutrient content of nixtamalized maize flour from South African maize is currently being conducted and is ongoing. The results will be published shortly in a separate publication as the current article has a different aim and scope. |
An important part of nixtamalization is the use of energy to carry it out, this is also important to consider when talking about expanding this type of food. The question is: where will the users get the energy from? Do they have money for this purpose? Is it sustainable? |
Thank you for spotting this fact. We agree with your statement and the research is also currently being conducted. It is ethnographical research that takes more time to analyze. However, the results will be published in a separate manuscript and a different journal. |
It is recommended to make a more specific conclusion on the evaluated aspects, without the need to fall into making a summary of the results obtained speaking quantitatively. |
The conclusion was re-written to be shorter and more specific. |
Reviewer comment: Otherwise, I consider it to be an interesting and very complete work, which should address the previous recommendations with the objective that the reader can take advantage of the information in an easy and clear way. |
Reviewer 4 Report
Comments and Suggestions for Authors
The objectives of the manuscript are clearly articulated, providing a strong foundation for the study. However, the current version requires a more explicit examination of how the chemical composition of the samples influenced the outcomes. The manuscript offers limited information regarding the chemical reactions that occurred during the analysis, which ultimately contributed to the promising results presented by the authors.
While the statistical analysis is adequate, there are several opportunities for enhancement. Expanding the descriptive statistics for all sample groups would provide a clearer overview of the data distribution and variability. Additionally, incorporating factorial ANOVA or MANOVA could explore potential interaction effects between variables such as chemical composition and processing conditions. Integrating effect sizes alongside p-values would also offer a better understanding of the practical significance of the findings.
To further strengthen the analysis, applying appropriate post-hoc tests and discussing their results would clarify specific differences between groups. Moreover, incorporating Principal Component Analysis (PCA) or other multivariate techniques could significantly enhance the understanding of how sample composition influences the observed preferences and outcomes. If the study includes measurements over time, time-series analysis or mixed-effects models could reveal trends not apparent in cross-sectional data. Finally, correlation and regression analyses could add predictive power and provide deeper insights into the relationships between variables.
Author Response
Comment: The objectives of the manuscript are clearly articulated, providing a strong foundation for the study. However, the current version requires a more explicit examination of how the chemical composition of the samples influenced the outcomes. The manuscript offers limited information regarding the chemical reactions that occurred during the analysis, which ultimately contributed to the promising results presented by the authors. While the statistical analysis is adequate, there are several opportunities for enhancement.
Please see the reviewers comments and auhor's reponses in the table below:
Expanding the descriptive statistics for all sample groups would provide a clearer overview of the data distribution and variability. Additionally, incorporating factorial ANOVA or MANOVA could explore potential interaction effects between variables such as chemical composition and processing conditions. Integrating effect sizes alongside p-values would also offer a better understanding of the practical significance of the findings. To further strengthen the analysis, applying appropriate post-hoc tests and discussing their results would clarify specific differences between groups. |
Thank you for the suggestion. ANOVA was done for the nixtamalized maize samples as there were enough samples. However, in this article we only show the averages of the results and compare them to nutrient content information that was sourced from websites. It would not be possible to do ANOVA or MANOVA. The complete nutritional analysis is ongoing and the interaction between proceeding conditions such as raw maize, the non-and nixtamalized flour and the produced products are being conducted. Those results are not ready yet but should be available later. However, the purpose of this study was to show that a product could be produced in a basic kitchen that is comparable to similar commercial products. |
Moreover, incorporating Principal Component Analysis (PCA) or other multivariate techniques could significantly enhance the understanding of how sample composition influences the observed preferences and outcomes. |
Principal Component Analysis were done for each of the products. However, that added another four graphs and their discussions. The authors took the decision to omit the PCA’s because the JAR tables, and particularly the mean drop and penalty data clearly showed the results and although further elucidation of the outcomes is interesting, the length of the manuscript and the number of tables and graphs is also considered by journals when publishing research. |
If the study includes measurements over time, time-series analysis or mixed-effects models could reveal trends not apparent in cross-sectional data. |
These are all excellent suggestions that future research could include. |
Finally, correlation and regression analyses could add predictive power and provide deeper insights into the relationships between variables. |
Correlations were done for the sensory results. However, in the interest of the length of the article, it was omitted. The reviewer is correct in all the suggestions in enhancing the results of the sensory and nutrition analysis. Such results will be published in due time. However, it was not the objective of this study. |
Round 2
Reviewer 1 Report
Comments and Suggestions for Authors
There are also some weaknesses that need to be addressed and some of the problems have not be corrected in the manuscript.
1. Line 426-427: “It ranged from 1.2 (maize nugget sample appearance and chickpea nugget mouthfeel) to 2.0 (chickpea sample taste).”
The penalty of potato nugget sample taste was 2.0, not chickpea sample taste (1.9).
2. Table 7, Table 8 and Table 9: suggest to express the data of % as point, such as 0.5, 4.4. The data in the tables should be consistent with the interpretation in the text.
3. Line 239-240: JAR analysis, suggest to add the evaluation criterion and sentences of the four sensory attributes for each product (nuggets, burgers, original chips and flavoured chips).
4. Line 70-71: “Govender (2014) identified the consumption of pap as a significant problem, as it does not provide enough nutrients to nourish the body”, suggest to add superscript 16 at the end of the sentence.
Author Response
1. Line 426-427: “It ranged from 1.2 (maize nugget sample appearance and chickpea nugget mouthfeel) to 2.0 (chickpea sample taste).”
The penalty of potato nugget sample taste was 2.0, not chickpea sample taste (1.9). |
Thank you for picking up this mistake. It was changed to “It ranged from 1.2 (maize nugget sample appearance and chickpea nugget mouthfeel) to 2.0 (potato sample taste). |
2. Table 7, Table 8 and Table 9: suggest to express the data of % as point, such as 0.5, 4.4. The data in the tables should be consistent with the interpretation in the text. |
I apologize; for some reason, the change did not save. I agree with it and make the changes again. |
3. Line 239-240: JAR analysis, suggest to add the evaluation criterion and sentences of the four sensory attributes for each product (nuggets, burgers, original chips and flavoured chips). |
The authors decided to add this information as a table since it would be impossible to describe in a paragraph. This information may assist other researchers in understanding and conceptualizing their own descriptors for JAR evaluations. It may increase the citations to this article; therefore, we agree that it should be included. |
4. Line 70-71: “Govender (2014) identified the consumption of pap as a significant problem, as it does not provide enough nutrients to nourish the body”, suggest to add superscript 16 at the end of the sentence. |
The reference was added at the end of the sentence. |

Reviewer 2 Report
Comments and Suggestions for Authors
The article submitted for review concerns the developing an acceptable nixtamalized maize product for South African consumers. The manuscript submitted for review is a full article on the development a consumer-acceptable nixtamalized maize product by employing consumerled methods that could be produced in a household kitchen from whole dried maize kernels. I would like to thank the Authors for introducing modifications, corrections and improvements in the current version of the manuscript. The current version of the manuscript looks much better and it may be accepted for further stages of publication.
Author Response
Comment: The article submitted for review concerns the developing an acceptable nixtamalized maize product for South African consumers. The manuscript submitted for review is a full article on the development a consumer-acceptable nixtamalized maize product by employing consumerled methods that could be produced in a household kitchen from whole dried maize kernels. I would like to thank the Authors for introducing modifications, corrections and improvements in the current version of the manuscript. The current version of the manuscript looks much better and it may be accepted for further stages of publication.
Response: We are grateful for the comments and the time and effort put into reviewing the manuscript.